# *Euglena* Attenuates High-Fat-Diet-Induced Obesity and Especially Glucose Intolerance

**DOI:** 10.3390/nu16213780

**Published:** 2024-11-04

**Authors:** Tengteng Ji, Bing Fang, Yutong Jin, Chenyan Zheng, Xinlei Yuan, Jianguo Dong, Le Cheng, Fang Wu

**Affiliations:** 1Key Laboratory of Precision Nutrition and Food Quality, Department of Nutrition and Health, China Agricultural University, Beijing 100193, China; 2College of Food Engineering and Biotechnology, Tianjin University of Science and Technology, Tianjin 300453, China; yuanxinlei@mail.tust.edu.cn

**Keywords:** *Euglena*, anti-obesity, weight loss, fat deposition

## Abstract

**Background:** Obesity, a global disease, can lead to different chronic diseases and a series of social health problems. Lifestyle changes, especially dietary changes, are the most effective way to treat obesity. *Euglena*, a novel food, has attracted much attention. Previous studies have shown that *Euglena* is an important modulator of the host immune response. In this study, the effects of *Euglena* as a nutritional intervention in high-fat-diet-induced obese C57BL/6J mice were investigated regarding adipose tissue accumulation and lipid and glucose metabolism by gavage at the dose of 100 mg/kg bodyweight for 9 weeks. This study is one of the few to investigate, in detail, the preventive effects of dietary *Euglena* on obesity. **Methods:** Five-week-old male C57BL/6J mice were fed with a high-fat diet (HFD) to induce obesity. An obesity model was created by feeding the high-fat diet for a period of 10 weeks. Obese mice were randomized into 2 groups with the same mean body weight, and no significant differences were observed between the groups: (1) the mice in the HEG group were maintained on a high-fat diet and daily gavaged with *Euglena* (100 mg/kg body weight) dissolved in saline (*n* = 7); and (2) the mice in the HFD group were maintained on a high-fat diet and daily gavaged with saline with the same volume (*n* = 7). The experiment finished after a nine-week period. **Results:** The results showed that *Euglena* could reduce the accumulation of white body fat, including subcutaneous fat and visceral fat, and mainly targeted subcutaneous fat. *Euglena* also reduced adipocyte particle size expansion, promoted lipolysis in adipose (adipose triglyceride lipase and hormone-sensitive triglyceride lipase) and liver tissue (reduced non-esterified fatty acid content), and improved obesity-induced ectopic fat deposition and glucose tolerance. **Conclusions:** Our findings suggest that *Euglena*, as a nutritional intervention in HFDs, efficiently reduces body weight and white adipose tissue deposition. The mechanism of *Euglena* is mainly though enhancing lipolysis. It is worth noting that *Euglena* β-glucan recovers the hyperglycemia and accumulation of ectopic fat within the liver induced by HFD. Our study is one of the few studies to report in detail the preventive effects of dietary *Euglena* on obesity in vivo. This study revealed that *Euglena* also has an important ameliorative effect on obesity and metabolic disorders, which laid a theoretical foundation for its future application in functional foods.

## 1. Introduction

Obesity is accompanied by excessive accumulation of body fat, leading to various abnormal health conditions and triggering metabolic syndrome [1]. Several studies have revealed that obesity triggers cardiovascular disease, type 2 diabetes mellitus (T2DM), and other chronic diseases [2,3,4]. It is well known that weight loss reverses metabolic abnormalities in individuals with T2DM, resulting in improved glycemic control. A weight loss of 15% or more can have a disease-modifying effect in patients with T2DM, which is unachievable with any other glucose-lowering intervention [5]. Therefore, weight loss through lifestyle changes, especially diet changes, is strongly recommended for obese individuals. Some studies have demonstrated that dietary fiber represents one of the most efficacious options for the improvement in obesity, which can not only reduce body weight, but improve metabolic disturbance by decreasing postprandial plasma glucose elevation [6,7,8].

*Euglena* (*Euglena gracilis*) contains abundant β-glucan [9], which is a dietary fiber commonly found in grains and algae in nature. *Euglena* is a motile unicellular alga and eukaryotic photosynthetic microorganism. It has secondary chloroplasts and is a mixotroph able to feed by photosynthesis or phagocytosis. While it has been reported that *Euglena* contributes to suppressing blood sugar levels [10], it has many functional components, including vitamins, minerals, amino acids, unsaturated fatty acids, and β-glucan. It was reported that β-glucans from grains (e.g., oatmeal) can reduce the risk of obesity, diabetes, and cardiovascular disease [11,12,13]. However, studies of *Euglena* mainly focused on immunomodulation, which can promote the production of antibodies in the body and improve immunity [14]. Regarding the anti-obesity or metabolism-modulated activity, only Sugimoto et al. reported that *Euglena* inhibits adipocyte differentiation based on an in vitro study using human adipose-derived stem cells [15]. The very few animal studies found that *Euglena* supplementation reduced serum low-density lipoprotein cholesterol (LDL-C) levels in obese mice, but did not have a beneficial effect on body weight [16]. This may be due to the way the supplement is pre-mixed in the feed, whereas positive anti-obesity results were achieved in oat [17] and yeast [18] with β-glucan supplemented by oral gavage. Considering *Euglena* has been widely applied as a novel food, as well as a recommended dose of dietary fiber, we evaluated the effects of *Euglena* as a nutritional intervention for the obese population due to its abundant β-glucan. In this study, we supplemented *Euglena* in obese mice, along with a high-fat diet, for 9 weeks, and evaluated fat deposition and glycolipid metabolism after the intervention. The dose studied here was based on the dose of β-glucan derived from whole grain (10%) at the recommended daily uptake of 100 g (100 g is based on the daily intake of whole grains per person in the Balanced Diet Pagoda for Chinese residents). This study provides in vivo evidence for the anti-obesity activity of *Euglena* and a theoretical support for the application of *Euglena* in functional food for the obese population to lose weight and improve their metabolic status.

## 2. Materials and Methods

### 2.1. Animals and Experimental Design

All the experiments were approved by the Animal Care and Ethics Committee of China Agricultural University (AW21403202-5-2). Five-week-old male C57BL/6J mice were purchased from Beijing Vital River Laboratory Animal Technology Co., Ltd., Beijing, China, and after one week’s acclimatization, mice were fed with a high-fat diet (HFD) to induce obesity. An obesity model was created by feeding the high-fat diet for a period of 10 weeks. Those weighing more than 15% above the control group were included in the follow-up study. A total of 14 mice were included in the follow-up experiment. Our animal experiments were designed according to the “3R principle”, which statistically requires a minimum of 6 mice per group. Obese mice were randomized into 2 groups with the same mean body weight, and no significant differences were observed between the groups (we computer generated random numbers to group animals indiscriminately at random): (1) the mice in the HEG (fed with high-fat diet and gavaged with *Euglena* β-glucan) group were maintained on a high-fat diet and daily gavaged with *Euglena* (100 mg/kg body weight, Beijing Zaochen Biotechnology Co., Ltd., Beijing, China) dissolved in saline (*n* = 7); and (2) the mice in the HFD group were maintained on a high-fat diet and daily gavaged with saline with the same volume (*n* = 7). The experiment finished after a nine-week period (The experiment was concluded when the HEG group exhibited a 10% reduction in weight relative to the HFD group). Mice of the same age and fed with a chow diet were used as negative controls (CON) at the end of the experiments (*n* = 8). To ensure consistency, we gavaged the mice at the same time each day. All cages were situated in the same room and on the same floor of the animal house, allowing for a uniform external environment. All experimental animals were maintained in individual cages (one animal per cage) at room temperature (24 ± 2 °C) with a 12 h light and dark cycle and free access to food and water. Details of the diets are shown in Table 1. Body weight and food intake were monitored on a weekly basis. All mice were euthanized by CO_2_ inhalation and cervical dislocation at the end of the experiments. White adipose tissue, including epididymis (eWAT), perirenal (pWAT), inguinal (ingWAT), and axillary (aWAT) tissue, and brown adipose tissue (BAT), as well as the blood and liver, were rapidly collected from the mice.

### 2.2. Body Composition Analysis and MRI

Body composition analysis and MRI were undertaken before euthanasia. Mice were examined for fat and lean mass using a body composition analysis and imaging system for awake small animals (NIUMAG, Suzhou, China, MesoQMR23-060H-I). The mice were then subjected to MRI to further characterize the two-dimensional spatial distribution of adipose tissue. Mice were anesthetized with 2.0% isoflurane and maintained at 36 ± 1 °C throughout the imaging procedure.

### 2.3. Histological Analysis and Oil Red O Staining

The tissues were fixed in 4% paraformaldehyde for a period of 24 h and embedded in paraffin. Sections from three mice for each group (3 μm) were stained with hematoxylin and eosin (H&E) (Solarbio, Beijing, China, #G1100, #G1080). Images were obtained using a Leica DM6B upright microscope and three views were randomly collected per section for analysis. The diameters of adipocytes were quantified using Image-Pro Plus 6.0 software (Media Cybernetics, Rockville, MD, USA). For lipid quantification in the liver, sections (10 μm) were stained with Oil Red O solution (Solarbio, #G1260).

### 2.4. Biochemical Analysis

Contents of triglycerides and NEFA (non-esterified fatty acid) in eWAT, ingWAT, and the liver were detected (after euthanasia) by colorimetric kits (Nanjing Jianjian Bioengineering Institute, #A110-1-1, Nanjing, China. Boxbio, Beijing, China, #AKFA008M), following the manufacturer’s protocols.

### 2.5. Gene Expression Analysis

Total RNA of eWAT, ingWAT, and the liver was extracted by TRIzol (Invitrogen, Waltham, MA, USA, #15596026); then, 1500 ng RNA was reverse-transcribed into cDNA using All-In-One 5X RT Mastermix (Abm, New York, NY, USA, #G492). Gene expressions were analyzed using real-time quantitative PCR with PowerUp™ SYBR™ Green (Thermo, Waltham, MA, USA, #A25742), and the expression was normalized to that of glyceraldehyde-3-phosphate dehydrogenase (*GAPDH*). The sequences of the primers used here are shown in Table 2.

### 2.6. Blood Lipid Profile

Serum contents of total triglycerides (TGs), cholesterol (TC), high-density lipoprotein cholesterol (HDL-C), and low-density lipoprotein cholesterol (LDL-C) were determined (after euthanasia) using an automated biochemical analyzer (Mindray, Shenzhen, China, BS-430).

### 2.7. Glucose Tolerance Test

Upon conclusion of the intervention, mice were fasted for 14 h and then intraperitoneally injected with a solution of glucose at the concentration of 1 g/kg body weight. Subsequently, we used a glucometer (Roche, ACCU-CHEK Performa, Mannheim, Germany) to measure the blood glucose content at 0, 15, 30, 60, 90, 120, and 180 min (the experiment ended when blood glucose returned to fasting levels). The area under the curve (AUC) was then calculated [19].

### 2.8. Statistical Analysis

The data are expressed as the mean ± SEM between groups. Figures were drawn by GraphPad Prism version 10.2.3 software. One-way ANOVA followed by Dunnett’s test was performed and significance was defined as a *p*-value less than 0.05 (indicated by * or #), 0.01 (indicated by ** or ##), or 0.001 (indicated by *** or ###).

## 3. Results

### 3.1. Euglena Reduces Body Weight and Fat Gain in Obese Mice

Throughout the experiment, the body weights of the mice fed HFD were consistently greater than those fed regular diets. The differences were significant (Figure 1A). By contrast, after *Euglena* gavage, the body weights of the HEG group increased more slowly than those of the HFD group. Significant differences in body weight between the two groups were observed from the fourth week onward (*p* < 0.05, Figure 1A). Body weight gain also significantly differed between different diets (CON vs. HFD and HFG) and between the HFD and HEG groups at the fourth week (*p* < 0.05, Figure 1B). Upon conclusion of the intervention, the terminal mean body weight of mice in the HEG group was 5.43 g less than that of mice in the HFD group (*p* < 0.05, Figure 1C), but still significantly higher than in the CON group (*p* < 0.05). Meanwhile, the spatial distribution of adipose tissue in the mice was examined using MRI (Figure 1D), which were consistent with the trend of terminal body weight. We further examined the fat and lean mass of awake mice, and the results showed that the lean mass of the mice from HFD and HEG groups decreased significantly at the end of the experiment, and the fat mass increased significantly (*p* < 0.01, Figure 1E,F). In addition, when assessing adipose tissue deposition, compared with CON mice, HFD-fed mice showed a remarkable increase in the percentages of sWAT (subcutaneous white adipose tissue, including ingWAT and aWAT), vWAT (visceral white adipose tissue, including eWAT, pWAT, and mWAT), and BAT (Figure 1H). When *Euglena* β-glucan was used for gavage, the percentages of sWAT and vWAT were both significantly reduced compare with those in the HFD group (Figure 1H).

### 3.2. Euglena β-Glucan Decreases Lipid Droplet Size in Adipocytes

H&E-stained sections showed that adipocyte diameters were smaller in CON mice and significantly larger in HFD-fed mice, and were decided by the size of lipid droplets in the cells. After intervention with *Euglena* β-glucan, the size of adipocytes in eWAT and ingWAT was significantly reduced (Figure 2). Moreover, the ratios of adipocytes with diameters in excess of 100 μm decreased from 19.40% and 15.02% (in HFD) to 8.62% and 5.25% (in HEG) in eWAT (*p* < 0.001, Figure 2D) and ingWAT (*p* < 0.001, Figure 2F), respectively.

### 3.3. Euglena β-Glucan Reduces Ectopic Fat Deposited in the Liver

We detected excessive lipid deposition in the liver of obese mice by oil red O staining (Figure 3A). According to the calculation of the staining area, it was found that ectopic fat in livers was reduced by 20.06% after *Euglena* β-glucan intervention (*p* < 0.001, Figure 3B). In addition, the significantly increased triglyceride and NEFA levels induced by HFD were relieved by *Euglena* β-glucan (*p* < 0.001, Figure 3C,D). We also measured the relative mRNA expression of genes involved in fatty acid de novo synthesis. In comparison with the CON, HFD significantly upregulated the mRNA expression of hepatic *Srebf1* (sterol regulatory element binding transcription factor 1), *Fasn* (fatty acid synthase), and *Scd1* (stearoyl-coA desaturase 1) (Figure 3E), which can be downregulated by *Euglena* β-glucan intervention (Figure 3E).

### 3.4. Euglena β-Glucan Recovers HFD-Induced Hyperglycemia but Not Improve Hyperlipidemia

Obesity is always companied with disturbed glucose metabolism. As shown in Figure 4, mice in the HFD group exhibited much higher fasting blood glucose (*p* < 0.05, Figure 4B) and significant glucose intolerance (Figure 4A,C). After *Euglena* β-glucan intervention, the fasting blood glucose of obese mice decreased significantly (*p* < 0.01, Figure 4B) and even returned to the level of the CON group. The impaired glucose tolerance induced by HFD also recovered to the control status (*p* < 0.01, Figure 4C). Regarding serum lipid profiles, after HFD feeding, serum contents of TC and LDL-C increased significantly, which could not be improved by *Euglena* β-glucan (*p* < 0.05, Figure 4D; *p* < 0.01, Figure 4F). Although serum content of HDL-C increased significantly after *Euglena* β-glucan intervention (*p* < 0.001, Figure 4F), no notable enhancements were observed in the ratios of LDL-C and HDL-C (*p* > 0.05, Figure 4E), or in the atherogenic index (*p* > 0.05, Figure 4G).

### 3.5. Euglena β-Glucan Enhances Lipolysis in Adipose Tissue

High levels of TG and NEFA were detected both in the eWAT and ingWAT of obese mice in the HFD group (*p* < 0.001), and intervention with *Euglena* significantly reduced TG and NEFA levels in ingWAT, whereas only NEFA levels were decreased in eWAT (*p* < 0.01, Figure 5A,B). This finding is consistent with the degree of changes in the percentage of adipose tissue at different anatomical locations shown in Figure 1H. Expression of lipolysis-related genes suggested that *Euglena* upregulated the expression of adipose triglyceride lipase (*Atgl*) and hormone-sensitive lipase (*Hsl*) (Figure 5C,D). In addition, adipogenesis was reduced to some extent, as demonstrated by the ability of *Euglena* to significantly inhibit the expression of acetyl-coenzyme A carboxylase alpha (*ACC*) (Figure 5C,D).

## 4. Discussion

It is estimated that over 2 billion individuals worldwide are currently classified as overweight or obese [20,21,22,23]. Obesity profoundly impacts the quality of life and livelihoods, even in people who appear to be healthy. There is substantial evidence that overweight and obesity and their comorbidities increase cardiovascular disease, overall morbidity, and mortality rates [24,25,26,27,28]. Although diet and exercise are the most efficacious methods for the treatment of obesity, dietary supplements with anti-obese activity attract great attention. *Euglena* has been approved as a novel food in countries including China, Japan, and USA due to its immunomodulatory effects [29,30,31,32,33,34], as well as being a source of dietary fiber. Different from oat, barley, or yeast, very few studies have reported the impacts of *Euglena* on glucose and lipid metabolism. Although Sugimoto et al. found it can inhibit adipocyte differentiation [15], Aoe et al. found it could only reduce serum LDL-C levels in obese mice, without reducing body weight [16]. This may be due to the relatively low efficient uptake dose when supplemented though the pre-mixed feed. In our study, *Euglena* was supplemented though oral gavage and at a dose between 2.5% and 5%, as used in Aoe et al. We found significant anti-obese activity, not only in terms of body weight and weight gain, but also in the proportions of adipose tissue deposited in the body (Figure 1). The obesity process was significantly blocked after only 4 weeks of intervention with *Euglena*. Upon conclusion of the intervention, both the MRI of the mice and the weight of adipose tissue suggested the anti-obesity activity of *Euglena*, especially targeting the sWAT (Figure 1).

As the largest endocrine and immune organ with high plasticity, adipose tissue plays a pivotal role in metabolism homeostasis [35]. Adipose tissue volume is determined by a combination of adipocyte number and size. Increased fat storage in fully differentiated adipocytes, leading to increased adipocyte size, is well documented and is considered the most significant mechanism for increased fat storage in adults [36,37]. It has been reported in the literature that there is a direct correlation between adipocyte size and number and fat mass in terms of sex and the site of deposition [38]. In the process of diet-induced obesity, adipocytes hold surplus energy by growing in size and number, which is reflected in the findings (Figure 2A). Adipose tissue is also highly plastic, capable of tissue expansion during overnutrition and contraction during undernutrition, and altering to a more metabolically favorable peripheral distribution under the influence of multiple factors [39]. For example, in the present study, after intervention with *Euglena*, adipocytes responded to energy changes by decreasing their excessive enlargement (Figure 2).

Moreover, the benefits of *Euglena* also include improvements in obesity-induced hyperglycemia and hyperlipidemia. An excessive lipid load in individuals with obesity leads to the excessive accumulation of triglycerides, which disrupts the dynamic balance between lipid synthesis and catabolism, leading to the ectopic deposits of fat in other organs, especially the liver, causing metabolic disorders and dyslipidemia, including increased TG and NEFA levels [40,41]. Therefore, we measured contents of TG and NEFA in adipose tissue and the liver. *Euglena* intervention significantly reduced TG and NEFA levels in sWAT and the liver, and, in vWAT, only NEFA levels tended to decrease (Figure 3C,D and Figure 5A,B). This finding further suggests that *Euglena* primarily targets sWAT, consistent with the deposition changes described above. The effect on lipid accumulation in the liver, on the other hand, may be achieved by modulating fatty acid synthesis from the outset. Using a glucose tolerance test, we found that *Euglena* efficiently cured high fasting blood glucose and glucose intolerance induced by HFD. This finding is consistent with that of Mo et al. [17]. Regarding the lipid-lowering effect, we examined four lipid profiles (TC, TGs, HDL-C, and LDL-C), cholesterol ratios (LDL-C/HDL-C), and the atherosclerotic index (AI: (TC-HDL)/HDL) [42,43]. HDL-C has long been regarded to be beneficial to cardiovascular health [44]. Our results demonstrated that although *Euglena* β-glucan significantly increased the serum concentration of HDL-C, the risk of atherosclerosis may not be efficiently improved (Figure 4E).

Furthermore, the plasticity of adipose tissue during obesity is characterized as the body’s energetic metabolic response, i.e., increased lipogenesis (energy storage) and decreased lipolysis (energy consumption) [39]. This link was improved by the intervention of *Euglena*, accompanied by the hydrolysis of TG, with a massive mobilization of lipids in white adipose lipid droplets, promoting lipolysis and the release of NEFA [45]. The process accelerated the loss of adipose tissue weight and reduced the buildup of lipid droplets of the liver (Figure 1E and Figure 4C). qPCR results also demonstrated an increase in the expression of lipolysis-related genes after *Euglena* intervention (*Atgl*, *Hsl*). Among them, *Euglena* promoted lipolysis in sWAT to a greater extent than in vWAT (Figure 3A,B). The process of lipolysis is dependent upon the inhibitory phosphorylation of *plin1*, a protein on the surface of lipid droplets. In the basis metabolic or anabolic state, *plin1* binds to comparative gene identification-58 (CGI-58). When stimulated by lipolytic signals, plin1 is phosphorylated, triggering the revitalization of adipose triglyceride lipase (*Atgl*). The revitalized *Atgl* then travels to the surface of the lipid droplet to hydrate TAG. pKA can also phosphorylate and activate *Hsl*, which links to *plin1* and facilitates the hydrolyzation of diacylglycerol to monoacylglycerol [45]. However, in our study, *plin1* did not change, possibly because the hypermetabolic state of the mice was excessively high after continuous high-fat feeding; thus, the *Euglena* intervention only exerted a weak effect. In addition, it has been reported in the literature that *plin1* has a bidirectional regulatory effect on lipid degradation [46,47]. For example, in the fasting or exercising state, *plin1* is highly phosphorylated by G protein-coupled receptors, which increases the affinity of lipase for the surface of lipid droplets and promotes lipolysis. *Euglena* β-glucan not only improves lipolysis in the body, but also decreases lipogenesis to some extent. This effect is manifested by the ability of *Euglena* to significantly inhibit the key enzyme for lipid de novo synthesis, *ACC* (Figure 5C,D), which has been widely reported as the rate-restricting enzyme for FA synthetic [48].

## 5. Conclusions

Our findings suggest that *Euglena*, as a nutritional intervention in HFDs, efficiently reduces body weight and white adipose tissue deposition. The mechanism of *Euglena* is mainly though enhancing lipolysis. It is worth noting that *Euglena* β-glucan recovers the hyperglycemia and accumulation of ectopic fat within the liver induced by HFD. Our study is one of the few studies to report in detail the preventive effects of dietary *Euglena* on obesity in vivo. This study revealed that *Euglena* also has an important ameliorative effect on obesity and metabolic disorders, which laid a theoretical foundation for its future application in functional foods. As evidenced by the results, *Euglena* β-glucan primarily targets sWAT. It is therefore pertinent to inquire why the fats in different anatomical locations respond differently to dextran, which may in part serve as a target for future obesity interventions. It is also important to identify whether there are other anti-obesity and/or hypoglycemic bioactive components in *Euglena*, as well as the dose–effect relationship based on more doses.

## Figures and Tables

**Figure 1 nutrients-16-03780-f001:**
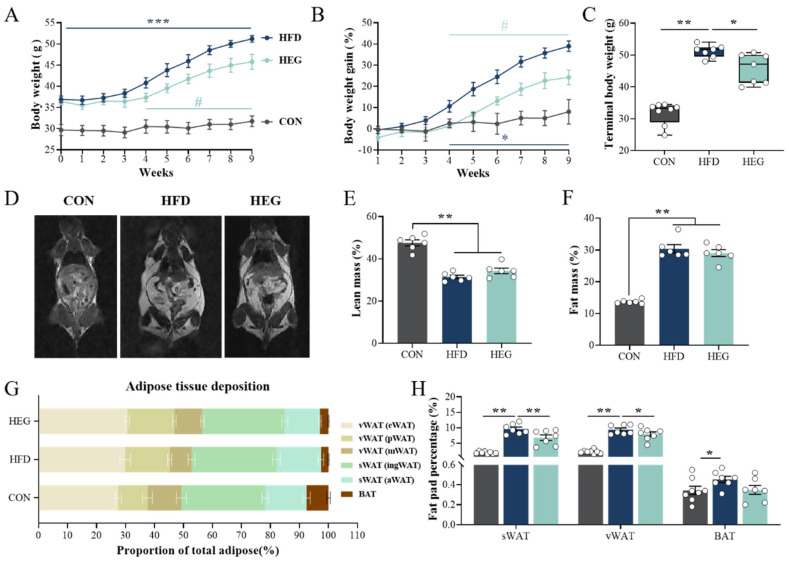
*Euglena* β-glucan reduces fat deposition and body weight induced by HFD. (**A**) Body weight and (**B**) body weight gain during the 9 weeks’ intervention. (**C**) The terminal body weight of mice in CON, HFD and HEG groups. (**D**) Representative MRI figures of mice in CON, HFD, and HEG groups. (**E**) Lean mass and (**F**) fat mass measured by MRI. (**G**) The deposition of adipose tissue in the types of eWAT, pWAT, mWAT, ingWAT, aWAT, and BAT. (**H**) vWAT, sWAT, and BAT of mice as a proportion of body weight in the CON, HFD, and HEG groups (CON: *n* = 8, HFD and HEG: *n* = 7). The values are presented as the mean ± SEM of seven or eight independent samples. * *p* < 0.05, ** *p* < 0.01 and *** *p* < 0.001 vs. the CON group; ^#^
*p* < 0.05 vs. the HFD group. Abbreviations: aWAT, axillary white adipose tissue; BAT, brown adipose tissue; CON, control chow fat; eWAT, epididymis white adipose tissue; HFD, high-fat diet and gavaged with saline; HEG, high-fat diet and gavaged with *Euglena* dissolved in saline; ingWAT, inguinal white adipose tissue; mWAT, mesenteric white adipose tissue; pWAT, perirenal white adipose tissue; sWAT, subcutaneous adipose tissue; vWAT, visceral adipose tissue.

**Figure 2 nutrients-16-03780-f002:**
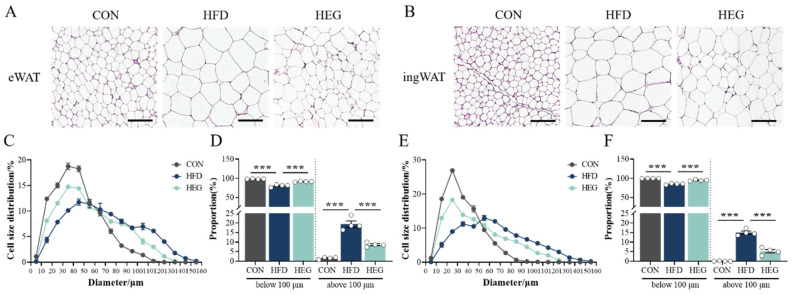
*Euglena* β-glucan alleviates HFD-induced hypertrophy in both vWAT and sWAT. (**A**,**B**) Representative images of H&E staining of eWAT (**A**) and ingWAT (**B**) (magnification: 20×; scale bars: 0.1 mm). (**C**,**E**) The distribution of adipocyte diameters and (**D**,**F**) ratios of adipocytes with diameters over 100 μm in eWAT (**C**,**D**) and ingWAT (**E**,**F**) (*n* = 4). Adipocyte size was determined using Image-Pro Plus software and, for each group, four mice and six independent fields per section were calculated. The values are presented as the mean ± SEM. *** *p* < 0.001. Abbreviations: eWAT, epididymis white adipose tissue; HE, hematoxylin and eosin; ingWAT, inguinal white adipose tissue.

**Figure 3 nutrients-16-03780-f003:**
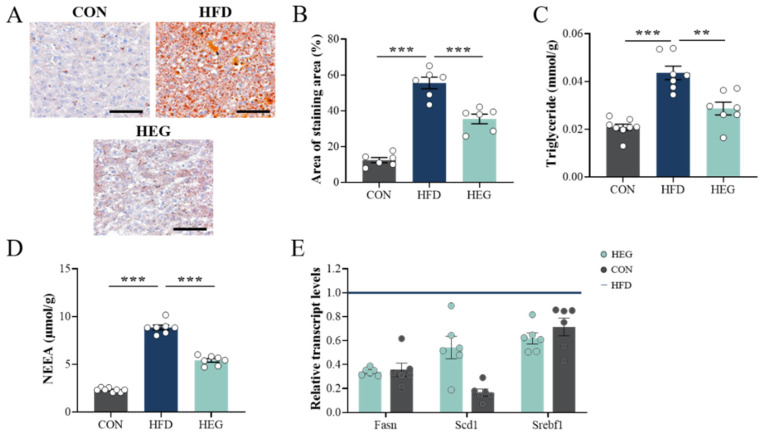
*Euglena* β-glucan alleviates HFD-induced excess lipid accumulation in the liver. (**A**) Representative H&E staining images of the liver from three groups (magnification: 10×; scale bars: 0.1 mm). (**B**) The ratio of the oil red O-stained area determined by Image-Pro Plus software (*n* = 6). (**C**) TG and (**D**) NEFA contents in the liver (CON: *n* = 8, HFD and HEG: *n* = 7). (**E**) Relative expression of genes involved in fatty acid de novo synthesis (*n* = 6). The values are presented as the mean ± SEM. ** *p* < 0.01, *** *p* < 0.001. Abbreviations: CON, control chow fat; HE, hematoxylin and eosin; HFD, high-fat diet and gavaged with saline; HEG, high-fat diet and gavaged with *Euglena* dissolved in saline; NEFA, non-esterified fatty acid; TG, triglycerides.

**Figure 4 nutrients-16-03780-f004:**
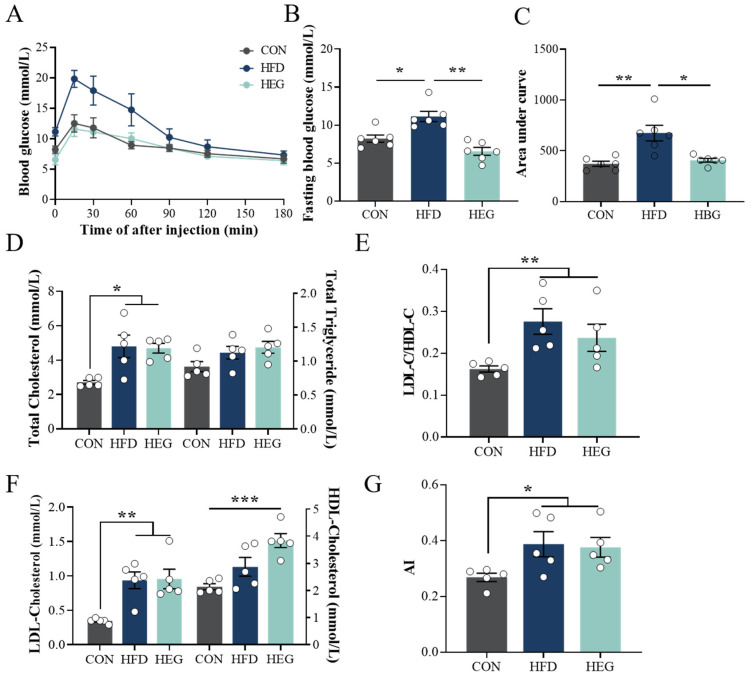
Effects of *Euglena* β-glucan on HFD-induced hyperglycemia and hyperlipidemia. (**A**) IPGTTs after 9 weeks. (**B**) Fasting blood glucose. (**C**) Area under the baseline curve in the IPGTT graph (*n* = 6). (**D**) TC and TG levels in the serum. (**E**) The ratio of LDL-C to HDL-C. (**F**) LDL-C and HDL-C levels in the serum. (**G**) Atherosclerosis index (AI) calculated by the ratio of TC minus HDL-C to HDL-C. The values are presented as the mean ± SEM (*n* = 5). * *p* < 0.05, ** *p* < 0.01, *** *p* < 0.001. Abbreviations: HDL-C, high-density lipoprotein cholesterol; IPGTT, intraperitoneal glucose tolerance test; LDL-C, low-density lipoprotein cholesterol; TC, total cholesterol; TG, triglycerides.

**Figure 5 nutrients-16-03780-f005:**
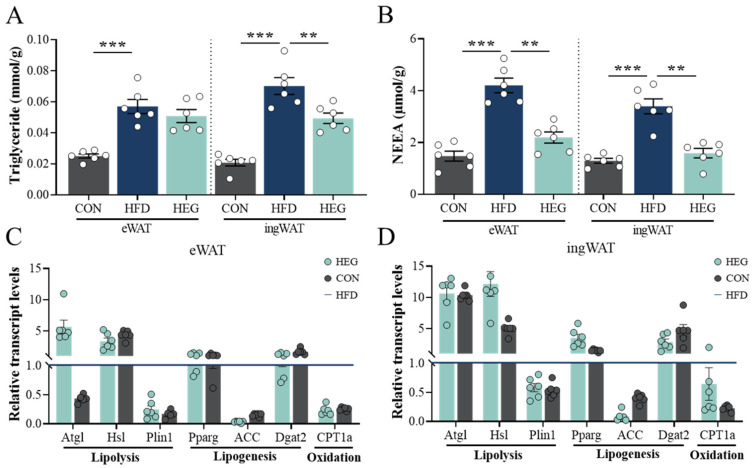
*Euglena* β-glucan enhances the lipolysis of hypertrophic adipocytes. (**A**) TG and (**B**) NEFA contents in eWAT and ingWAT. (**C**,**D**) Relative mRNA expression of genes involved in lipolysis (e.g., *Atg1*, *Hsl*, *Plin1*), lipogenesis (e.g., *PPARg*, *ACC*, *Dgat2*), and oxidation (*CPT1a*) in (**C**) eWAT and (**D**) ingWAT measured by qRT-PCR. The values are presented as the mean ± SEM (*n* = 6). ** *p* < 0.01, *** *p* < 0.001. Abbreviations: *ACC*, acetyl-coenzyme A carboxylase alpha; *Atgl*, adipose triglyceride lipase; *CPT1a*, carnitine O-palmitoyltransferase 1; *Dgat2*, diacylglycerol acyltransferase 2; eWAT, epididymis white adipose tissue; *Hsl*, hormone-sensitive triglyceride lipase; ingWAT, inguinal white adipose tissue; mRNA: messenger RNA; NEFA, non-esterified fatty acid; *Plin1*, perilipin 1; *PPARg*, peroxisome proliferator activated receptor-g; TG, triglycerides.

**Table 1 nutrients-16-03780-t001:** The composition of the diet used in this study.

Diet	CON	HFD		HEG	
Company	Research Diets (Beijing, China)	Research Diets (New Brunswick, NJ, USA)		Research Diets (New Brunswick, NJ, USA)	
Catalog	AIN-93G	D12492		D12492	
Cal density (kcal/g)	3.72	5.24		5.24	
Fat (%)	11	60		60	
	soybean oil/2.8%	soybean oil/5.5%	Lard/5.5%	soybean oil/5.5%	Lard/5.5%
Protein (%)	34	20		20	
Carbohydrate (%)	55	20		20	

Abbreviations: CON, control chow fat; HFD, high-fat diet, gavaged with saline; HEG, high-fat diet, gavaged with *Euglena* dissolved in saline.

**Table 2 nutrients-16-03780-t002:** List of primer sequences.

Primers		Sequence (Direction: 5′ to 3′)
*ACC*	Forward	GATGAACCATCTCCGTTGGC
	Reverse	CCCAATTATGAATCGGGAGTGC
*Atgl*	Forward	CTGAGAATCACCATTCCCACATC
	Reverse	CACAGCATGTAAGGGGGAGA
*CPT1a*	Forward	CACTGCAGCTCGCACATTAC
	Reverse	CCAGCACAAAGTTGCAGGAC
*Dgat2*	Forward	GCGCTACTTCCGAGACTACTT
	Reverse	GGGCCTTATGCCAGGAAACT
*Fasn*	Forward	CACAGTGCTCAAAGGACATGCC
	Reverse	CACCAGGTGTAGTGCCTTCCTC
*GAPDH*	Forward	TGTGTCCGTCGTGGATCTGA
	Reverse	CCTGCTTCACCACCTTCTTGA
*Hsl*	Forward	TCCTCAGAGACCTCCGACTG
	Reverse	ACACACTCCTGCGCATAGAC
*Plin1*	Forward	CAAGCACCTCTGACAAGGTTC
	Reverse	GTTGGCGGCATATTCTGCTG
*PPARg*	Forward	GGAAGACCACTCGCATTCCTT
	Reverse	TCGCACTTTGGTATTCTTGGAG
*Scd1*	Forward	GCAAGCTCTACACCTGCCTCTT
	Reverse	CGTGCCTTGTAAGTTCTGTGGC
*Srebf1*	Forward	CGACTACATCCGCTTCTTGCAG
	Reverse	CCTCCATAGACACATCTGTGCC

Abbreviations: *ACC*, acetyl-coenzyme A carboxylase alpha; *Atgl*, adipose triglyceride lipase; *CPT1a*, carnitine O-palmitoyltransferase 1; *Dgat2*, diacylglycerol acyltransferase 2; *Fasn*, fatty acid synthase; *GAPDH*, glyceraldehyde-3-phosphate dehydrogenase; *Hsl*, hormone-sensitive triglyceride lipase; *Plin1*, perilipin 1; *PPARg*, peroxisome proliferator activated receptor-g; *Scd1*, stearoyl-coA desaturase 1; *Srebf1*, sterol regulatory element binding transcription factor 1.

## Data Availability

The authors confirm that the data supporting the findings of this study are available within the article.

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
