# Peer review of "Euglena Attenuates High-Fat-Diet-Induced Obesity and Especially Glucose Intolerance"

_nutrients, 2024, doi:10.3390/nu16213780_

Round 1
Reviewer 1 Report
Comments and Suggestions for Authors
Euglena attenuates high-fat diet induced obesity and especially glucose intolerance
Type: article
The article deals with a very interesting topic, namely the effect of eugenyl beta glucan in relation to obesity. The topic of obesity is a relatively current topic, because obesity is growing worldwide, and with it also associated diseases.
The article is has clearly defined goals, which are answered in the article.
I suggest to write down part 2.1 more specifically.
Line 63 – after one weeks acclimatization, mice were fed with a HFD to induce obesity. And after this week, obese mice were divided into 2 groups - HEG and HFD. My question is, HEG mice were daily gavaged only with euglena? Or HFD together with euglena? And HFD mice were gavaged with saline and HFD? Because line 72 continue: During the 9th week intervention, mice were contiued to feed with HFD. It means that experiment lasted 9 weeks, what is described aso in results. please give a more specific description.
Results and Discussion is clear and brief.
Please, in Table 1, add one more column with composition of HEG (or HEG together with HFD?)
Author Response
- The article deals with a very interesting topic, namely the effect of euglena beta glucan in relation to obesity. The topic of obesity is a relatively current topic, because obesity is growing worldwide, and with it also associated diseases.
The article is has clearly defined goals, which are answered in the article.
I suggest to write down part 2.1 more specifically.
Response: We're so grateful that you took the time to read our article and share your valuable insights. We've made some detailed updates in response to your feedback on part 2.1. Here are the details:
All the experiments were approved by the Animal Care and Ethics Committee of China Agricultural University (AW21403202-5-2). Five-week-old male C57BL/6J mice were purchased from Beijing Vital River Laboratory Animal Technology Co., Ltd., and after one week’s acclimatization, mice were fed with a high-fat diet (HFD) to induce obesity. An obesity model was created by feeding the high-fat diet for a period of 10 weeks. Those weighing more than 15% above the control group were included in the follow-up study. A total of 14 mice were included in the follow-up experiment. Our animal experiments are designed according to the “3R principle”, which statistically requires a minimum of 6 mice per group. Obese mice were randomized into 2 groups with the same mean body weight and there were no significant differences between groups (we computer-generated random numbers to group animals indiscriminately at random): (1) The mice in the HEG group were maintained on a high-fat diet and daily gavaged with euglena (100 mg/kg body weight, Beijing Zaochen Biotechnology Co., Ltd) dissolved in saline (n=7); and (2) the mice in the HFD group were maintained on a high-fat diet and daily gavaged with saline with the same volume (n=7). The experiment finished after a nine-week period. And the mice with the same age and feed with a chow diet were used as negative controls (CON) at the end of the experiments (n=8). To ensure consistency, we gavage the mice at the same time each day. All cages are situated in the same room and on the same floor of the animal house, allowing for a uniform external environment. All experimental animals were housed in single cages at room temperature (24±2°C) with a 12-hour light anlined dark cycle and free access to food and water. Details for the diets were shown in Table 1. Body weight and food intake were monitored every week. All mice were euthanized by CO2 inhalation and cervical dislocation at the end of the experiments. White adipose tissue, including epididymis (eWAT), perirenal (pWAT), inguinal (ingWAT), axillary (aWAT), and brown adipose tissue (BAT), as well as blood and liver were rapidly collected from the mice.
Furthermore, the aforementioned information has been incorporated into the manuscript (part 2.1, lines 72-97).
- Line 63 - after one week acclimatization, mice were fed with a HFD to induce obesity. And after this week, obese mice were divided into 2 groups - HEG and HFD. My question is, HEG mice were daily gavaged only with euglena? Or HFD together with euglena? And HFD mice were gavaged with saline and HFD? Because line 72 continue: During the 9th week intervention, mice were continued to feed with HFD. It means that experiment lasted 9 weeks, what is described also in results. please give a more specific description.
Response: We extend our sincerest apologies for any distress caused by our failure to provide sufficient clarity. We would like to provide you with a detailed explanation: (1) The mice in the HEG group were maintained on a high-fat diet and daily gavaged with euglena (100 mg/kg body weight, Beijing Zaochen Biotechnology Co., Ltd) dissolved in saline (n=7); and (2) the mice in the HFD group were maintained on a high-fat diet and daily gavaged with saline with the same volume (n=7). The experiment finished after a nine-week period. Furthermore, the missing information was added in the revised manuscript (lines 82-87).
- Please, in Table 1, add one more column with composition of HEG (or HEG together with HFD?)
Response: Thank you very much for your professional questions. We have added a column to Table 1 to provide additional descriptions of the dietary components of the HEG group. Here are the details:
Table 1. The composition of the diet used in this study.
|
Diet |
CON |
HFD |
HEG |
||
|
Company |
Research Diets |
Research Diets |
Research Diets |
||
|
Catalog# |
AIN-93G |
D12492 |
D12492 |
||
|
Cal density (kcal/g) |
3.72 |
5.24 |
5.24 |
||
|
Fat (%) |
11 |
60 |
60 |
||
|
soybean oil/2.8% |
soybean oil/5.5% |
Lard/5.5% |
soybean oil/5.5% |
Lard/5.5% |
|
|
Protein (%) |
34 |
20 |
20 |
||
|
Carbohydrate (%) |
55 |
20 |
20 |
||
Furthermore, the aforementioned information has been incorporated into the manuscript (lines 99-102).
Reviewer 2 Report
Comments and Suggestions for Authors
Thank you for the opportunity to review this manuscript.
It presents an interesting study about the effects of euglena as a nutritional intervention in high-fat-diet induced obese mice.
However, I have some concerns presented below.
General comments
Please, include the tables and figures in the main text, near the place they are signaled in it.
Please, include in the legends of the tables and figures the explanation of the acronyms you have used on it.
Please, make the use of the acronyms fit with the Journal recommendations. For example, you are using HEG mice in line 69. It is the first time it appears in the text and you are not explaining the meaning. There are more examples; such as NEFA (line 98). GAPDH, line 106. sWAT and vWAT, line 140. Srebf1, Fasn and Scd1, line 159. Lines 236-261, review the entire paragraph for the acronyms.
Specific Comments
Abstract
Please, avoid the use of the acronyms Atgl, Hsl and NEFA in the abstract.
Introdution
Could you add and explanation about botanical definition of the Euglena, please?
Line 53. Please clarify: “The dose studied here was based on the dose of β-glucan derived from whole grain (10%) at the recommended daily uptake of 100 g.” Which is the reference for this dose? How is it possible that you stated in the abstract that you used 100 mg/kg and here you are speaking about a dose of 100 g?
2. Materials and Methods
Line 63: “after one week’s acclimatization, mice were fed with a high-fat diet (HFD) to induce obesity. Those weighing more than 15% above the control group were included in the follow-up study.” Thus, you were feeding with the high-fat diet only the 14 mice divided afterward in the supplemented and not supplemented groups. Please, clarify.
In this paragraph, lines 63-74, please clarify the issue of the groups, and the issue of the dosage and origin of the euglena.
2.2. Body composition analysis and MRI. We suppose you made this analysis just before euthanasia? When we get results section, we see that you have measured this each week, but I think you have to clarify here. And was it the same for 2.4. Biochemical analysis and 2.6. Blood lipid profile? Please, clarify.
3. Results
Line 137. “the mice fed an HFD decreased”. Please clarify about which groups and what moment of assessment are you talking about.
Line 149. “decreased from 19.40% and 15.02% to 8.62% and 5.25% in eWAT 149 (p<0.001, Figure 2D) and ingWAT (p<0.001, Figure 2F), respectively” Please clarify: “decreased from 19.40% and 15.02% in HFD to 8.62% and 5.25% in HEG…”
4. Discussion
Line 185. “There are more than 2 billion people globally are still overweight or obese [19].” Please, syntax review.
Please, add the limitations here at the end of the discussion.
Author Response
General comments
- Please, include the tables and figures in the main text, near the place they are signaled in it.
Response: Thank you very much for your professional questions. Following what you said, we have included the tables and figures in the main text, near the place they are signaled in it (Table 1: line 99-102; Table 2: lines 131-138; Graphical abstract: lines 27-28; Figure 1: lines 177-190; Figure 2: lines 198-206; Figure 3: lines 218-226; Figure 4: lines 239-246; Figure 5: lines 258-267).
- Please, include in the legends of the tables and figures the explanation of the acronyms you have used on it.
Response: Thank you very much for your professional opinion. In light of your remarks, we have included in the legends of the tables and figures the explanation of the acronyms we have used on it (lines 27-28, 99-102, 131-138, 177-190, 198-206, 218-226, 239-246, 258-267).
- Please, make the use of the acronyms fit with the Journal recommendations. For example, you are using HEG mice in line 69. It is the first time it appears in the text and you are not explaining the meaning. There are more examples; such as NEFA (line 98). GAPDH, line 106. sWAT and vWAT, line 140. Srebf1, Fasn and Scd1, line 159. Lines 236-261, review the entire paragraph for the acronyms.
Response: We extend our sincerest apologies for any distress caused by our failure to provide sufficient clarity. In light of your remarks, the acronyms that appear for the first time in the text were subjected to close scrutiny and their meanings were elucidated. And the abbreviations included in the tables and figures, further clarifications have been provided (lines 21-23, 56, 83, 85, 119, 128, 172-173, 215-216).
Specific Comments
Abstract
- Please, avoid the use of the acronyms Atgl, Hsl and NEFA in the abstract.
Response: Thank you for your professional guidance. We have replaced all the acronyms in the summary (lines 21-23).
Introdution
- Could you add and explanation about botanical definition of the Euglena, please?
Response: Euglena (Euglena gracilis), one of the microalgae, has long been known to produce various vitamins and amino acids essential for human health [1]. Euglena is a motile unicellular alga and eukaryotic photosynthetic microorganism. It has secondary chloroplasts and is a mixotroph able to feed by photosynthesis or phagocytosis. While it has been reported that Euglena contributes to suppressing blood sugar levels [2]. It has many functional components, including vitamins, minerals, amino acids, unsaturated fatty acids, and β-glucan. We re-described this part in the manuscript (lines 44-49).
References:
[1] Aemiro, A.; Watanabe, S.; Suzuki, K.; Hanada, M.; Umetsu, K.; Nishida, T. Effect of substituting soybean meal with euglena (Euglena gracilis) on methane emission and nitrogen efficiency in sheep. Anim. Sci. J. 2019, 90 (1), 71-80.
[2] Isegawa, Y., Activation of immune and antiviral effects by euglena extracts: a review. Foods 2023, 12 (24), 15.
- Line 53. Please clarify: “The dose studied here was based on the dose of β-glucan derived from whole grain (10%) at the recommended daily uptake of 100 g.” Which is the reference for this dose? How is it possible that you stated in the abstract that you used 100 mg/kg and here you are speaking about a dose of 100 g?
Response: We apologize for the lack of clarity in our presentation. The 100g mentioned here is based on the daily intake of whole grains per person in the Balanced Diet Pagoda for Chinese residents. The 100mg/kg in the abstract is our gavage dose for experimental animals. We re-described this part in the manuscript (lines 65-66). Thank you for your comments!
Materials and Methods
- Line 63: “after one week’s acclimatization, mice were fed with a high-fat diet (HFD) to induce obesity. Those weighing more than 15% above the control group were included in the follow-up study.” Thus, you were feeding with the high-fat diet only the 14 mice divided afterward in the supplemented and not supplemented groups. Please, clarify.
Response: I offer my sincere apologies if our formulation has been unclear. The manuscript has been augmented with additional material (lines 76-78). Here are the details: An obesity model was created by feeding the high-fat diet for a period of 10 weeks. Those weighing more than 15% above the control group were included in the follow-up study. A total of 14 mice were included in the follow-up experiment.
- In this paragraph, lines 63-74, please clarify the issue of the groups, and the issue of the dosage and origin of the euglena.
Response: Thank you very much for your professional opinion. In light of your remarks, we have clarified the issue of the groups, and the issue of the dosage and origin of the euglena in the manuscript (lines 76-88). Here are the details: An obesity model was created by feeding the high-fat diet for a period of 10 weeks. Those weighing more than 15% above the control group were included in the follow-up study. A total of 14 mice were included in the follow-up experiment. Obese mice were randomized into 2 groups with the same mean body weight and there were no significant differences between groups (we computer-generated random numbers to group animals indiscriminately at random): (1) The mice in the HEG group were maintained on a high-fat diet and daily gavaged with euglena (100 mg/kg body weight, Beijing Zaochen Biotechnology Co., Ltd) dissolved in saline (n=7); and (2) the mice in the HFD group were maintained on a high-fat diet and daily gavaged with saline with the same volume (n=7). The experiment finished after a nine-week period. And the mice with the same age and feed with a chow diet were used as negative controls (CON) at the end of the experiments (n=8).
- 2.2. Body composition analysis and MRI. We suppose you made this analysis just before euthanasia? When we get results section, we see that you have measured this each week, but I think you have to clarify here. And was it the same for 2.4. Biochemical analysis and 2.6. Blood lipid profile? Please, clarify.
Response: Thank you very much for your professional opinion. Body composition analysis and MRI were made before euthanasia. Biochemical analyses were performed after euthanasia. We performed 2.4 and 2.6 by collecting serum, adipose tissue and liver samples. The above is also explained in the corresponding section of the manuscript (lines 104, 120, 142).
Results
- Line 137. “the mice fed an HFD decreased”. Please clarify about which groups and what moment of assessment are you talking about.
Response: I offer my sincere apologies if our formulation has been unclear. The manuscript has been augmented with additional material (lines 168-170). Here are the details: We further examined the fat and lean mass of awake mice, and the results showed that the lean mass of the mice from HFD and HEG group decreased significantly at the end of the experiment, and the fat mass increased significantly.
- Line 149. “decreased from 19.40% and 15.02% to 8.62% and 5.25% in eWAT 149 (p<0.001, Figure 2D) and ingWAT (p<0.001, Figure 2F), respectively” Please clarify: “decreased from 19.40% and 15.02% in HFD to 8.62% and 5.25% in HEG…”
Response: I offer my sincere apologies if our formulation has been unclear. We have added the corresponding groups (lines 196-197). Here are the details: Moreover, the ratios of adipocytes with diameters excess 100 μm decreased from 19.40% and 15.02% (in HFD) to 8.62% and 5.25% (in HEG) in eWAT (p<0.001, Figure 2D) and ingWAT (p<0.001, Figure 2F), respectively.
Discussion
- Line 185. “There are more than 2 billion people globally are still overweight or obese [19].” Please, syntax review.
Response: Thank you very much for your professional opinion. We reviewed the relevant literature and added to the manuscript (Ref 20-23).
- Please, add the limitations here at the end of the discussion.
Response: Thank you very much for your professional opinion. We have added to the limitations in the manuscript (lines 358-360). Here are the details: And it is important to identified weather there were other anti-obesity and/or hypoglycemic bioactive components in euglena, as well as the dose-effect relationship based on more doses.
Round 2
Reviewer 2 Report
Comments and Suggestions for Authors
I want to thank the authors for all the corrections performed to clarify the manuscript. Only one more observation: Line 83: "The mice in the HEG group were maintained on a high-fat diet ". Please, explain here HEG, it is the first time it appears in the text. Congratulations on your manuscript.